# A Method to Compare the Delivery of Psychiatric Care for People with Treatment-Resistant Schizophrenia

**DOI:** 10.3390/ijerph17207527

**Published:** 2020-10-16

**Authors:** Anna Alonso-Solís, Susana Ochoa, Eva Grasa, Katya Rubinstein, Asaf Caspi, Kinga Farkas, Zsolt Unoka, Judith Usall, Elena Huerta-Ramos, Matti Isohanni, Jussi Seppälä, Elisenda Reixach, Jesús Berdún, Iluminada Corripio, m-RESIST Group

**Affiliations:** 1Psychiatry Department, Institutd’ Investigació Biomèdica-Sant Pau (IIB-SANT PAU), Hospital de la Santa CreuiSant Pau; Universitat Autònoma de Barcelona (UAB), 08193 Barcelona, Spain; aalonsos@santpau.cat (A.A.-S.); egrasa@santpau.cat (E.G.); ICorripio@santpau.cat (I.C.); 2CIBERSAM, Biomedical Research Networking Centre Consortium on Mental Health, 28029 Madrid, Spain; sochoa@pssjd.org (S.O.); mehuertas@pssjd.org (E.H.-R.); 3Parc Sanitari Sant Joan de Déu, 08830 Sant Boi Llobregat, Barcelona, Spain; 4The Gertner Institute of Epidemiology and Health Policy Research, Sheba Medical Center, Tel-Aviv University, 6997801 Tel Aviv, Israel; rubins.katya@gmail.com (K.R.); acaspi@013net.net (A.C.); 5Department of Psychiatry and Psychotherapy, Semmelweis University, 1083 Budapest, Hungary; farkas.kinga@med.semmelweis-univ.hu (K.F.); unoka.zsolt@med.semmelweis-univ.hu (Z.U.); 6Centre for Life Course Health Research, University of Oulu, 90570 Oulu, Finland; matti.isohanni@oulu.fi (M.I.); Jussi.Seppala@esshp.fi (J.S.); 7Department of Psychiatry, Oulu University Hospital, 90220 Oulu, Finland; 8South Carelia Social and Health Care District, Psychiatric and Substance Use Services, 53130 Lappeenranta, Finland; 9TicSalut Health Department, Generalitat de Catalunya 08005 Barcelona, Spain; ereixach@ticsalutsocial.cat (E.R.); jberdun@ticsalutsocial.cat (J.B.)

**Keywords:** treatment-resistant schizophrenia, information and communication technologies (ICT), mental health services, Europe, mHealth

## Abstract

Introduction: Community services are gaining ground when it comes to attention to patients with psychiatric diseases. Regarding patients with treatment-resistant schizophrenia (TRS), the use of information and communication technology (ICT) could help to shift the focus from hospital-centered attention to community services. This study compares the differences in mental health services provided for patients with TRS in Budapest (Hungary), Tel-Aviv (Israel) and Catalonia (Spain) by means of a method for the quick appraisal of gaps among the three places, for a potential implementation of the same ICT tool in these regions. Methods: An adapted version of the Description and Standardised Evaluation of Services and Directories in Europe for Long Term Care (DESDE-LTC) instrument was made by researchers in Semmelweis University (Budapest, Hungary), Gertner Institute (Tel-Aviv, Israel) and Hospital de la Santa Creu I Sant Pau and Parc Sanitari Sant Joan de Déu (Catalonia, Spain). Results: Two types of outpatient care services were available in the three regions. Only one type of day-care facility was common in the whole study area. Two residential care services, one for acute and the other for non-acute patients were available in every region. Finally, two self-care and volunteer-care facilities were available in the three places. Conclusion: Although the availability of services was different in each region, most of the services provided were sufficiently similar to allow the implementation of the same ICT solution in the three places.

## 1. Introduction

The deinstitutionalization paradigm shifts the focus from hospital-centered attention to community services [1,2]. In European countries these changes tend to have similar objectives and principles of accessibility and equity. However, the implementation process is different in individual countries, shaping the everyday reality of patients differently [3]. Although the main trend has been to avoid and shorten hospital-centered care, the quality of inpatient care has also developed to include psychosocial elements and a longitudinal view and optimize antipsychotic medication [4].

A review by Becker and Kilian [5] showed that in Western Europe there has been an increase in community services with less in-patient treatment, albeit with substantial variability in psychiatric service systems within individual countries. In this sense, several studies have revealed differences in the pattern of care for people with schizophrenia, even though their clinical characteristics are similar [6,7]. The authors concluded that a description of mental health services across Europe is required. Similarly, another study demonstrated that a better coordination between system services could improve treatment and reduce the use of emergency psychiatric services [8].

Several studies have focused on the development of new methodologies for a structured evaluation of services considering their structure, organization and use [3,9,10,11,12,13]. The description of service patterns could be useful to assess and compare the availability of services in different countries and find ways to improve efficacy and optimize delivery. Recently, some studies have compared the availability of mental health services in several European countries by means of the Description and Standardised Evaluation of Services and Directories in Europe for Long Term Care (DESDE-LTC) instrument [10] reporting significant variations not only in care availability [14], but also in the typology and characteristics of these services across the study areas [15,16].

Treatment-resistant schizophrenia (TRS) is a severe form of schizophrenia and a frequent condition that psychiatrists worldwide have to deal with, as one-fifth to one-third of all patients with schizophrenia are considered to be resistant to treatment [17]. In the case of TRS patients, the use of services is varied. While a few of them seldom receive care, many others are frequent users receiving long-term care. In fact, several studies have demonstrated that patients who mainly use hospital services show the highest clinical severity and disability [18,19,20]. 

During the last years, the integration of information and communication technologies (ICTs) into the health care services has permitted the creation of interventions aimed to ameliorate the quality of life of psychiatric patients. In the case of TRS patients, ICT solutions could help overcome the principal barriers of the illness, by (i) offering tailored interventions; (ii) favoring continuity of care; (iii) promoting the empowerment of patients. The use of new ICT in the treatment of TRS could help to offer better care to these patients and thus reduce hospital admissions. A necessary first step, however, is to describe the network of services involved in caring for people with TRS. 

The present study is based on the framework of m-RESIST (mobile therapeutic attention for treatment-resistant schizophrenia), a European project developed by our group, designed to define and develop an ICT tool to provide continuous monitoring, access to care and empowerment of patients with TRS [21]. As three regions were involved in the study, Budapest, Tel-Aviv and Catalonia, the platform had to be designed in a way that it could be implemented in different regions with different regulations, different language interfaces and different health care professionals. Thus, during the first period of the m-RESIST project (definition of the solution), mental health care services in these three regions were identified and compared in order to discover and highlight the possible barriers of its future implementation.

The aim of the present study is to compare the differences in mental health services in TRS in Budapest, Tel-Aviv and Catalonia by means of a method for the quick appraisal of gaps among the three places.

## 2. Research Methods

### 2.1. Identifying Current Available Services

In order to identify the existing mental health services, the DESDE-LTC (Description and Standardised Evaluation of Services and Directories in Europe for Long-term Care) instrument has been used in three regions [11]. DESDE-LTC is an instrument for the standardized description and classification of services for long-term care in Europe (hospital and community). The aim of the instrument is to use a common operational definition across all types of health services (hospitalizations, community and rehabilitation services and self-help devices). In this line, the completion of the whole instrument provides a comprehensive mapping of the structure and level of service provision in a catchment area, and allows the comparison of services across different geographical areas. Section B of the instrument has been validated considering content validity, internal structure and practical usability by policy makers [18,22]. In our case, the questionnaire was used for a comparison of available mental health services in Budapest, Tel-Aviv and Catalonia. 

In order to provide a comprehensive and simplified mapping of structure and level of service provision in the three regions, Section B of the instrument was used. This section uses tree diagrams for the classification of services and international comparisons. This section of the DESDE-LTC instrument includes the following four tree diagrams of services:(1)Outpatient care services: these are facilities that (i) involve contact between staff and users for some purpose related to management of their clinical and social difficulties, and (ii) are not provided as a part of delivery of residential or day and structured activity services.(2)Day-care services: these are facilities which (i) are normally available to several users at a time (rather than delivering services to individuals one at a time); (ii) provide some combination of treatment for problems related to long-term care needs; (iii) have regular opening hours during which they are normally available; (iv) expect service users to stay at the facilities beyond the periods in which they have face-to-face contact with staff. The service is not simply based on individuals coming for appointments with staff and then leaving immediately after their appointments.(3)Residential care services: these services include facilities that provide beds overnight for users for a purpose related to the clinical and social management of their health condition, ranging from brief hospitalizations to long-term care.(4)Self-help and volunteer-care services: the aim of these facilities is to provide users with long-term care needs support, self-help or contact with unpaid staff. These facilities offer accessibility, information, and outpatient day and residential care.

The location of each service in the tree is identified by a combination of a letter and a number: a capital “O”, “D”, “R” or ”S”, which indicates whether the service is part of Outpatient care (O), Day care (D), Residential care (R) and/or Self-help care (S), and a number to identify each final branch within these trees.

An example of the DESDE-LTC tree diagram is shown in Figure 1. This figure is based on the outpatient care services “O” classification. Each branch contains information about the availability of services in the region for acute and non-acute scenarios, type of attention (mobile or non-mobile), hours of attention (limited or unlimited) and intensity of attention (high, medium or low). In the case of the Residential Care tree (R), branches are similar to O services. Day Care (D) branches include acute and non-acute services. In the case of non-acute services, these were categorized as work, work-related, non-work structure care and non-structure care. 

### 2.2. Procedure

In order to collect the information to complete each branch of the DESDE-LTC instrument, the DESDE-LTC was provided to researchers from Semmelweiss University (Budapest, Hungary), The Gertner Institute (Tel-Aviv, Israel), and Hospital de la Santa Creu I Sant Pau and Parc Sanitari Sant Joan de Déu (Catalonia, Spain). In the case of Hungary and Israel, as there were no governmental documents covering this information, all data were collected by an expert group of multidisciplinary professionals from each center that included psychiatrists, psychologists and nurses with clinical experience (including coordinators of services), and the research team involved in the project. The data collected were discussed by the clinicians and researchers and included information based on the common practice at Gertner Institute and Semmelweis and knowledge about other services available. In the case that some information was unknown, the researchers contacted the planners and directors of mental health services of their region. 

In the case of Catalonia, researchers were provided with an official catalogue validated by the Health Department of the Catalan Government with all available mental health services in the region of Catalonia. Similarly, to Israel and Hungary, in case of doubt, researchers contacted planners and directors of mental health services in Catalonia. Moreover, in Catalonia, we also had the benefit of a previously completed document with all the information contained in the DESDE-LTC [23].

Additionally, professionals involved in Section B of the DESDE-LTC in each region completed a questionnaire describing each service and the type of clinicians involved. Moreover, an estimated time dedicated to each patient in each service was performed.

The study ran from April 2015 to June 2015, so the information collected was based on services available in this period of time.

## 3. Results

### 3.1. Outpatient Care Services

Two types of outpatient care services were available in all three regions. The first one (O3.1), acute, non-mobile 24-h attention, was designed to provide health-related care including assessments and initial treatments in response to a crisis, deterioration in physical or mental state or behavior or social functioning related to the condition. The second facility available in all three areas provided continuous care (O8.1), including regular contact with a health professional, which may be long-term if required. Moreover, Budapest and Catalonia also had services that could provide acute, mobile and limited hours attention (O2.1) and non-acute services with mobile high intensity (O5.1) and non-mobile with medium (O9.1) and low-intensity contacts (O10.1). (Table 1).

Professionals involved in the services available in all three areas were psychiatrists and nurses, while psychologists and social workers were present only in Tel-Aviv and Catalonia. The visiting time spent by professionals was similar in the three regions (Table 2).

### 3.2. Day-Care Services

In relation to day-care facilities, there was one service available in all three regions (D4). It was a facility for non-acute patients which provided structured activities other than work and work-related care. Such activities may have included skills training, creative activities such as art or music and group work. These activities were designed to be available for at least 25% of opening hours. In all three regions, these facilities were available for service users who could attend for at least the equivalent of four half days per week. These facilities provided clinical long-term care (physical, psychological and/or social), and also offered structured activities related to social and cultural participation. Regarding acute services addressed to episodic treatment, Budapest and Catalonia had a similar service (D0.1), and in the case of continuous treatment, three regions had services with different degrees of intensity. Moreover, Budapest and Catalonia shared the availability of the D3 service, which was a work-related care service of high intensity (Table 1).

Regarding professionals involved in delivering the facility available in the three places (D4), some differences among regions emerged; while nurses were present in Budapest and Tel-Aviv, psychologists and social workers were present in Tel-Aviv and Catalonia (Table 2). The time spent in the professionals’ attention was similar in each region. 

### 3.3. Residential Care Services

As shown in Table 1, two facilities were available in all three regions, one for acute patients and the other one for non-acute patients in residential care services. In the first one (R1: with 24 h physician cover), users were admitted due to a deterioration of their physical or mental status severe enough to require continuous 24-h surveillance, and/or require special isolation measures. The second facility available in the three regions (R11: without 24-h physician cover) was addressed to non-acute patients. This facility provided residential care also during non-working hours but where there was a procedure that guaranteed users to receive 24-h care. In this type of facility, no fixed maximum period of residence was specified. Moreover, Budapest and Catalonia shared the availability of non-acute services with 24-h physician cover in a hospital setting (R4) and non-acute services without 24-h physician cover with limited time (R8).

Considering the professionals involved in the facilities available in the three regions, psychiatrists and nurses were present in all the regions in the R1 service, and psychiatrists, nurses, social workers and support workers in the R11 service. In both R1 and R11, attention time was the same. See Table 2.

### 3.4. Self-Help and Volunteer-Care Services

Two self-care and volunteer-care facilities were available in the three regions, both providing information on care. In the first place, facilities aimed at users with long-term care needs (S1.1), where graduate professionals providing assessment, interventions or support to users, were below 60% of the total full-time equivalent personnel. In the second place, the three regions had other facilities designed for users with long-term care where at least 60% of staff are graduate professionals trained or specifically qualified for providing assessments, interventions and support to users. In the case of Tel-Aviv and Catalonia, these two types of facilities, with both non-professional and professional staff, also provided accessibility to care (S1.2 and S2.2, respectively). Common devices were identified in Tel-Aviv and Catalonia, which aimed to provide accessibility to care (S2.2 and S1.2) and day-care services (S1.4) (Table 1).

In this branch, information regarding professional attention was not gathered due to the low presence of professionals in such services.

## 4. Discussion

The DESDE-LTC instrument has been useful for the comparison of available services in each region to treat people with TRS. Although the availability of services was different in each region, most of the services provided were sufficiently similar to allow the implementation of the same mHealth solution in the three regions. 

The implementation of an mHealth platform addressing TRS patients should take into account the services which were available in the three regions. These services were: outpatient resources with 24-h emergency access and outpatient care with high intensity, day centers and rehabilitation services, hospital and residential resources, and self-help resources. The regions included in the project had different social and cultural realities and the process of deinstitutionalization has been approached in different ways and at different times [24,25]. However, we have been encouraged to note that common services were available providing different types of mental health attention, which would allow continuous treatment and support the running of digital solutions. 

Outpatient care (acute and non-acute) facilities were available in the three regions, but mobile services were only available in Budapest and Catalonia. Case management services and mobile interventions addressing better community care were available in two of the three regions [26]. Moreover, previous research has suggested that outpatient mental health services have some difficulties in engaging severe patients [27]. As reported by our group, the use of forums, web information and greater availability of medical care were issues of concern regarding continuity of attention [28]. In this sense, the use of new technologies could help to better engage patients in community services.

Regarding day-care services, all three regions had acute and non-acute services, although their availability was different in each region. In general terms, the deinstitutionalization process moved towards a greater community and personal approach in order to meet daily needs [29]. These services offered long-term care and structured activities related to social contact. Social contact has been detected as one of the most important needs in people with schizophrenia [30] and TRS [28]. Moreover, the lack of social contact was closely related with higher levels of family burden and disability [31]. Using an mHealth platform, patients would be attended to in a more structured and continuous way in order to meet this specific need. Only Catalonia and Budapest had day-care services based on work activities. Taking this into account, work activities should be registered but not consider one of the core treatment aspects to be developed in the early stages of implementing an mHealth solution. 

In relation to residential care, all three regions had services for acute and non-acute patients. The use of acute services in TRS patients is high and more related to higher disability levels [18,19,20], as mentioned previously. However, new technologies could be useful to reduce the rate of re-hospitalization. Hence, reinforcing the network between community services and hospitalization services is a must in the treatment of TRS, as suggested by Bush and colleagues [32].

Finally, self-help services were usually managed by users or families with the guidance of professionals. In Budapest, these kinds of resources were usually independent of professional support. Although the idea is to use new technologies as a tool for the improvement of mental health services, the possibility to develop specific modules for use by non-professional associations should be considered. Moreover, the empowerment of people with schizophrenia should possibly integrate peer help in the development of new technologies, as other researchers have done for other problems [33].

The types of professionals involved in the treatment of people with schizophrenia in the three regions were similar, although some differences appeared. Professionals involved in the three regions for emergency services were psychiatrists, so the implementation of an mHealth solution in these services should consider the psychiatrist as the reference figure of attention. With regard to outpatient services not related to emergencies, other professionals such as nurses and psychologists were present in the three regions. In these kinds of services, interventions based on ICT tools addressed to psychological or health care interventions could be developed considering the availability of these professionals. As the day-care resources also include occupational therapists in their teams, the inclusion of everyday tasks could be implemented in these services. The use of multidisciplinary teams is necessary in order to improve results [34]. Along the same lines, different professionals could better assess all the important areas to be improved in people with schizophrenia, and new technologies should include repeated assessment of patients to better determine the needs to be met [35]. 

Regarding the time spent in each service by professionals, our results showed that the mean time is around 30 min per visit, oscillating between 5 and 60 min depending on the service and the professional. In the implementation of new technologies, we should take into account this limitation on the accessibility of services and the amount of time devoted to treat each patient in order to implement an ecological tool for the treatment of TRS.

Thanks to DESDE-LTC, this study provided a thorough comparison of available services in each region utilized to treat people with TRS. In this sense, one of the most important strengths of our study is that it allows us to compare the availability of mental health services in three different regions, permitting the extrapolation of this methodology to other mental health studies. However, it has its limitations, as this instrument has not been used in the same way in the three regions. While in Catalonia we had a recent catalogue describing all the available services, in Budapest and Tel-Aviv the availability of services was assessed by consulting professionals. Moreover, it should be considered that no objective quality control measures were performed in order to ensure the accuracy of the information provided by them; however, the completion was performed by an expert group of clinicians and researchers. It is important to note that results obtained in Catalonia, Budapest and Tel-Aviv might be difficult to extrapolate to Spain, Hungary or Israel, respectively, as there can be large differences in the delivery of health services among regions in the same country. This is particularly significant in the case Spain, where 17 different autonomous regions co-exist and have jurisdiction over their own health policies.

To conclude, this study compares the differences in the mental health services available in three regions (Budapest, Tel-Aviv and Catalonia) for the treatment of TRS patients. To this end, we used the DESDE-LTC instrument, a method for the quick appraisal of gaps in mental health services among different regions. We found that most of the services provided are sufficiently similar and with similar profile of professionals involved. In this sense, the m-RESIST ICT solution could be implemented under the same conditions in the mental health care environment of these three places. This is of special interest due to the fact that mHealth projects increasingly tend to have a global vision, involving several countries with differences in their mental health care systems.

## Figures and Tables

**Figure 1 ijerph-17-07527-f001:**
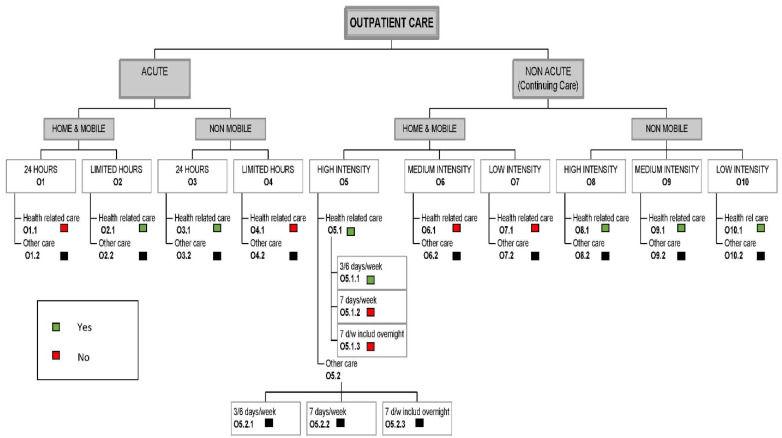
Description and Standardised Evaluation of Services and Directories in Europe for Long Term Care (DESDE-LTC) tree diagram of outpatient care services in Catalonia. Green squares: facility available; red squares: facility not available; black squares: facilities not evaluated as they refer to care unrelated to health.

**Table 1 ijerph-17-07527-t001:** Summary of the results.

				DESDE Code	Budapest (Hungary)	Tel-Aviv (Israel)	Catalonia (Spain)
**Outpatient Care (O)**	Acute	Home and Mobile	24 h	O1.1			
			Limited hours	O2.1	x		x
		Non-Mobile	24 h	**O3.1**	x	x	x
			Limited hours	O4.1	x		
	Non-Acute	Home and Mobile	High Intensity	O5.1	x		x
			Medium Intensity	O6.1			
			Low Intensity	O7.1	x		
		Non-Mobile	High Intensity	**O8.1**	x	x	x
			Medium Intensity	O9.1	x		x
			Low Intensity	O10.1	x		x
**Day Care (D)**	Acute	Episodic	High Intensity	D0.1	x		x
			Other Intensity	D0.2			
		Continuous	High Intensity	D1.1		x	x
			Other Intensity	D1.2	x		x
	Non-Acute	Work	High Intensity	D2	x		x
			Low Intensity	D6			
		Work-related care	High Intensity	D3	x		x
			Low Intensity	D7			
		Non-work structured care	High Intensity	**D4**	x	x	x
			Low Intensity	D8	x		
		Non-structured care	High Intensity	D5			
			Low Intensity	D9			
**Residential Care (R)**	Acute	24-h physician cover	Non-Hospital	R0			
			Hospital	**R1**	x	x	x
		Non-24-h physician cover	Non-Hospital	R3.1			
			Hospital	R3.0			
	Non-Acute	24-h physician cover	Non-Hospital	R5			x
			Hospital	R4	x		x
		Non-24-h physician cover	Time limited	R8	x		x
			Indefinite stay	**R11**	x	x	x
**Self-help and Volunteer Care (S)**	Professional staff		Information on care	**S2.1**	x	x	x
			Accessibility to care	S2.2		x	x
			Outpatient care	S2.3	x		
			Day care	S2.4			x
			Residential care	S2.5			
	Non-professional staff		Information on care	**S1.1**	x	x	x
			Accessibility to care	S1.2		x	x
			Outpatient care	S1.3	x		
			Day care	S1.4		x	x
			Residential care	S1.5			

DESDE-LTC codes in bold indicate the health care services which are common in the three regions. Acute: providing initial care and treatment in response to a crisis situation; Non-acute: providing users with regular contact with a health professional; Home and Mobile: the contact normally takes place in a wide range of locations including the user’s own home. At least 50% of the contacts are made outside the place where said service is established; Non-mobile: Facilities that do not meet the criteria for “mobile & home”; Episodic: Facilities in which care is usually provided to patients with deteriorating health status in a single or limited number of episodes and during a specified period of time. Continuous: the care is provided on an ongoing basis—non-episodic, at least 5 days a week for a limited period of time; Work: Work Facilities that provide users with the opportunity to be paid for their work; Work-related care: Facilities in which users carry out a work-related activity but where users are not paid or are paid less than 50% of the expected local salary for this work. Non-work structured care: Facilities that provide structured activities that are not work-related and that at least 25% of the working day would be available; Non-structured care: Facilities that meet the criteria for non-acute day service but where structured activities are not offered, the main functions of the service being the provision of social contact, practical help and/or support; Outpatient High Intensity: Facilities with the capacity to make face-to-face contact with users at least 3 times a week; Outpatient Medium Intensity: Facilities that can provide biweekly care when required; Outpatient Low Intensity: Facilities that do not have the capacity to serve their users on a biweekly basis; Outpatient Other intensity: Facilities that provide episodic acute care but do not meet criteria for high, medium or low intensity; Day Care Episodic High intensity: Facilities in which attention is routinely provided to patients with deterioration in their health status in a single or limited number of episodes of care and during a specified period of time; Day Care Continuous High Intensity: Admission to the service is usually made at less than 72 h; Day Care Non-acute High Intensity: Facilities that are normally available to serve users for at least the equivalent of 4 half days per week. Residential 24-h physician cover: Facilities located within hospitals where there is 24-h medical coverage.

**Table 2 ijerph-17-07527-t002:** Dedication time and professionals involved in common services.

DESDE Code	Professionals			Time Spent		
	Budapest (Hungary)	Tel-Aviv (Israel)	Catalonia (Spain)	Budapest (Hungary)	Tel-Aviv (Israel)	Catalonia (Spain)
O3	Psychiatrists	Psychiatrists	Psychiatrists	20–60 min.	20–60 min.	20–60 min.
	Nurses	Nurses	Nurses	20–50 min.	20–50 min.	20–50 min.
	-	Social workers	-	-	15–60 min.	-
O8	Nurses	-	Nurses	15 min.	-	30 min.
	-	Psychiatrist	Psychiatrist	-	60 min.	30 min.
	-	Psychologist	Psychologist	-	30–60 min.	45 min.
	-	Social workers	Social workers	-	20–50 min.	30 min.
D4	Nurses	Nurses	-	5–15 min.	20 min.	-
	-	Psychiatrist	-	-	20 min.	-
	-	Psychologist	Psychologist	-	30–50 min.	45 min
	-	Social Workers	Social workers	-	20–50 min.	30 min.
	-	-	Occupational therapist	-	-	60 min.
	-	-	Monitors	-	-	60 min.
R1	Psychiatrist	Psychiatrist	Psychiatrist	20 min–24 h	20 min–24 h	20 min–24 h
	Nurses	Nurses	Nurses	20 min–24 h	20 min–24 h	20 min–24 h
	-	Psychologist	Psychologist	-	30–50 min.	30–50 min.
	-	Social workers	-	-	20–30 min.	20–30 min.
R11	Psychiatrist	Psychiatrist	Psychiatrist	20–30 min.	20–30 min.	20–30 min.
	Nurses	Nurses	Nurses	30–60 min.	30–60 min.	30–60 min.
	Social workers	Social workers	Social workers	20–30 min.	20–30 min.	20–30 min.
	Support workers	Support workers	Support workers	24 h.	24 h.	24 h.
	-	-	Psychologist	-	-	45 min.

See footnote in Table 1 for the characteristics of the DESDE codes. We have not included information from the self-help and volunteer-care domain due to the fact that these type of services are self-help and voluntary and the proportion of professionals is lower than 60%.

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
