# Peer review of "A Method to Compare the Delivery of Psychiatric Care for People with Treatment-Resistant Schizophrenia"

_ijerph, 2020, doi:10.3390/ijerph17207527_

Round 1

Reviewer 1 Report

The authors have performed an interesting study, aiming to develop a method that will fill in the gaps in mental health services, with the upper aim of better care of patients with TRS and the reduction of hospitalizations.

Please find below some comments that, in my opinion, will improve the quality of the manuscript.

  • In the introduction part, the studies that are mentioned by authors have a big gap regarding the years of publication. I would suggest authors search more and also include studies that have performed in between.
  • Moreover, why the authors decided to focus on TRS? It is indeed a severe mental disorder, and I authors dedicated a paragraph on this, but the approach could be used for other more frequent disorders.
  • In my opinion, the instrument DESDE-LTC is not clearly described, and authors should be provide more information regarding this. It is also necessary to better describe Figure 1 in the text.
  • please also provide the duration of the study. The reference period is mentioned, but not the duration. And also, if the study was performed in 2015, why the authors delayed the publication? The reason I am mentioning this is that I am afraid that things might have changed during this 5-year period. At least this should be mentioned as a limitation of the study.
  • The part of the results needs to be elaborated more. Authors provide the two tables, but the accompanying text if not sufficient. Maybe a good idea would also be to start with the text and then to provide the tables, after reporting them in the text.
  • Also, I consider it important to provide an estimation of the number of patients that turn to these different services; maybe an additional column in table 2.

Author Response

Reviewer 1

The authors have performed an interesting study, aiming to develop a method that will fill in the gaps in mental health services, with the upper aim of better care of patients with TRS and the reduction of hospitalizations.

Please find below some comments that, in my opinion, will improve the quality of the manuscript.

In the introduction part, the studies that are mentioned by authors have a big gap regarding the years of publication. I would suggest authors search more and also include studies that have performed in between.

Answer: Thanks to reviewer 1 for this comment. We agree with the reviewer and we have added more recent references.

Moreover, why the authors decided to focus on TRS? It is indeed a severe mental disorder, and I authors dedicated a paragraph on this, but the approach could be used for other more frequent disorders.

Answer: This is a fair comment. The present project was performed under the framework of a major project (m-RESIST), focused on the application of an ICT tool for TRS patients. Although TRS is a frequent condition (around 30%of patients with schizophrenia) and involve a high burden for patients and society, we agree with the reviewer that the methodology of our study could also be applied to other mental health conditions.

In this regard, our study could be used as an example of quick appraisal of gaps in mental health services between different countries or health care systems.

In my opinion, the instrument DESDE-LTC is not clearly described, and authors should be provide more information regarding this. It is also necessary to better describe Figure 1 in the text.

Answer: Thanks to reviewer 1 for this comment. The DESDE-LTC instrument and Figure 1 have been better described in the manuscript.

Please also provide the duration of the study. The reference period is mentioned, but not the duration.

Answer: Thanks to reviewer 1 for this comment. We have added in the text the duration of the study. From April 2015 until June 2015

And also, if the study was performed in 2015, why the authors delayed the publication? The reason I am mentioning this is that I am afraid that things might have changed during this 5-year period. At least this should be mentioned as a limitation of the study.

Answer: This is a fair comment, and we agree with the reviewer that things might have changed during this 5-year period. However, our study can still be used as an example of developing a method for quick appraisal of gaps in mental health services. As mentioned above, we took advantage of the m-RESIST project and therefore our target population was TRS in Spain, Hungary and Israel. But as the reviewer pointed out, this same methodology can be used as an example by other researchers for quick appraisal of gaps in mental health services in other mental health conditions among different countries or health care systems.

In addition, as this was a secondary aim of the main project (m-RESIST), we decided to prioritize clinical outcomes and other papers of the project, delaying the writing of the present manuscript.

The part of the results needs to be elaborated more. Authors provide the two tables, but the accompanying text if not sufficient. Maybe a good idea would also be to start with the text and then to provide the tables, after reporting them in the text.

Answer: Thanks to reviewer 1 for this comment. We have added more text in the results section. In addition, following the reviewer suggestion, we have start the results section with the text, followed by the tables.

Also, I consider it important to provide an estimation of the number of patients that turn to these different services; maybe an additional column in table 2.

Answer: Thanks to reviewer 1 for this comment. We agree with the reviewer that this is very interesting information. Unfortunately, we did not collect this information.

Reviewer 2 Report

The work describes a method for appraisal of potential gaps in mental health services for treatment resistant schizophrenia, pointing to the description of the available  mental health services in three countries: Hungary, Israel and Spain. 

Methods and results are well described.

The major suggestion is to edit the language carefully, possibly by a native speaker. 

Author Response

Reviewer 2

The work describes a method for appraisal of potential gaps in mental health services for treatment resistant schizophrenia, pointing to the description of the available mental health services in three countries: Hungary, Israel and Spain. 

Methods and results are well described.

Answer: Thank you very much for this comment.

The major suggestion is to edit the language carefully, possibly by a native speaker. 

Answer: Thanks to reviewer 2 for this comment. The document has been thoroughly revised by a native English speaker and changes have been made throughout the manuscript.

Reviewer 3 Report

Title need to reflect that this work was done in community services not other setttings.
Introduction is long - summurize and focus on research problem.
Methods is clear. But authors did not discuss DESDE-LTC instrument validation and usability. Please incorporate one section on that.
Table 1 and Table 2 should have footnoot explaining terms.
Table 2 need to presented better in results.
Discussion lacks strenghts and weaknesses. No implications for further research.
Conclusion, need to reflect findings please rewrite.
Refrences 15% are 15 years old consider reniewing them.

Author Response

Reviewer 3

Title need to reflect that this work was done in community services not other setttings. 

Answer: Thanks to reviewer 3 for this comment. The project was performed considering all the services of mental health available. The residential section included information about admissions in non-acute and acute services. In this regard, a better description of the DESDE-instrument has been added in the method section of the manuscript in order to better understanding.

Introduction is long - summarize and focus on research problem.

Answer: This is a fair comment. The introduction section has been reduced. 

 Methods is clear. But authors did not discuss DESDE-LTC instrument validation and usability. Please incorporate one section on that. 

Answer: Thanks to reviewer 3 for this comment. A better explanation and references (including validation and usability) of the DEDSDE-LTC instrument has been provided in the text as follow

Table 1 and Table 2 should have footnoot explaining terms. 

Answer: We agree with reviewer 3. We have added footnotes to Tables 1 and 2.

Table 2 need to presented better in results. 

Answer: Thanks to reviewer 3 for this comment, which has also pointed out by reviewer 1. In this regard, a better explanation of the results have been provided in the text.

Discussion lacks strenghts and weaknesses. No implications for further research. 

Answer: We agree with the review that some strengths and weakness of the study should be comment. We have re-written this paragraph in the discussion section considering implications for further research:

Conclusion, need to reflect findings please rewrite. 

Answer: The conclusion has been re-written in order to describe the main findings of the project:

Refrences 15% are 15 years old consider reniewing them.

Answer: We have added more recent references.

Reviewer 4 Report

Thank you for the opportunity to review this manuscript, which describes use of the DESDE-LTC tool (in abbreviated form) and an additional questionnaire to compare services available in Hungary, Israel and Spain, and the resultant findings. There are some difference in emphasis between the stated aim (L111-113: “The aim of this study is to develop a method for quick appraisal of gaps in mental health services for treatment-resistant schizophrenia, with a practical application in Hungary and in Israel and having Catalonia (Spain) as the gold standard”) and the focus of the manuscript, which largely concentrates on the outcomes and is reflected in the manuscript title (“Differences in mental healthcare services in Hungary, Israel and Spain for people with treatment-resistant schizophrenia”)

I have two main concerns about the manuscript:

  1. For a paper whose stated aim is methodology development, there is very little in the manuscript about development of the methodology – essentially use of one part of an existing tool, and one additional questionnaire concerning the type and time commitment of clinicians involved in service provision. There does not appear to have been any attempt at validation either of the modified DESDE-LTC or of the additional instrument, as deployed here. While a large part of the methodology section describes the section of the DESDE-LTC that was used, there is minimal information on how it was used: merely that it was provided to researchers in the three countries, and where those researchers lacked information they contacted planners and directors of mental health services in their country. The researchers in Spain also had a previously completed DESDE-LTC as a resource.

    However, there is no information on how many researchers completed the questionnaire in each country, who they were (e.g. profession(s), seniority, clinical experience), the extent to which they needed to consult to acquire data in practice, whether there was any confirmation or cross-checking of the data etc. For example, from what is described, it appears possible that a single person who was confident in his/her knowledge (and therefore did not consult) but who was actually out of date or mistaken could have supplied multiple incorrect data points for a country without correction. For Hungary and Israel, the institutions named for the researchers in the methodology section appear to be primarily education and research institutions; had those researchers any direct, current clinical knowledge/experience?

    For the additional questionnaire, even less information has been provided. It is simply stated (L168-9) that, “Additionally each country completed a questionnaire describing, for each service, which clinicians were involved and the estimated time dedicated for them in an ordinary visit.” Was this completed by the same researchers? Was any consultation involved? Similar questions arise as for the DESDE-LTC component above.

  2. This leads to my second key concern, which is that data has been specified with a level of confidence and precision that seems very unlikely to be justifiable. This particularly the case for the data in Table 2 (the caption for which makes no mention of estimation). No objective basis for the personnel and time estimates has been provided, and it is unknown whether or how much consultation took place in their generation, yet specific time ranges (with an implied accuracy of minutes) are set out for multiple professions and settings.

    For the DESDE-LTC data it is stated that for Hungary and Israel, “as there are no governmental documents covering this information, and based on the knowledge that clinical practices are homogenous throughout both countries, all data were based on the common practice at Gertner Institute and Semmelweis.” It is not clear whether the same approach was taken for the personnel and time estimates but there is nothing to suggest otherwise.

    Even if it is accepted that all institutions offered the same range of services (and this seems implausible, given likely differences in size and budgets), it stretches credibility to assume that the same can be said of their specific operating systems to the extent that estimated patient care times for multiple professions (of unknown derivation) in one institution could be assumed to hold true nationally. In the section on limitations the possibility that data from Catalonia may not be representative of nationwide data is considered, but in fact this is true for all three countries and its likelihood places it beyond being a mere limitation; it is a significant methodological flaw that undermines the credibility of the data, especially those in Table 2. At best, they can be considered institutional estimates, and even then there appear to have been no objective quality control measures to provide any assurance of their accuracy.

Other comments:

Figure 1: The green and red squares have been defined as ‘yes’ and ‘no’ respectively; however there is no indication of what the black squares represent.

Table 2: Based on Table 1, there are additional services in the self-help and volunteer care domain that were common to all three locations: S1.1 and S2.1. These have not been included in Table 2, although in the discussion (L207-215) it is stated that both of these involve graduate professionals at proportions below and above 60% of total full time equivalent personnel respectively. Either they should have been included in Table 2 or their absence from it should have been explained.

From a language perspective, the manuscript is generally well written, with some minor lapses. Formatting also needs some attention, e.g. in several places, spaces between words appear to have been omitted.

Author Response

Reviewer 4

Thank you for the opportunity to review this manuscript, which describes use of the DESDE-LTC tool (in abbreviated form) and an additional questionnaire to compare services available in Hungary, Israel and Spain, and the resultant findings. There are some difference in emphasis between the stated aim (L111-113: “The aim of this study is to develop a method for quick appraisal of gaps in mental health services for treatment-resistant schizophrenia, with a practical application in Hungary and in Israel and having Catalonia (Spain) as the gold standard”) and the focus of the manuscript, which largely concentrates on the outcomes and is reflected in the manuscript title (“Differences in mental healthcare services in Hungary, Israel and Spain for people with treatment-resistant schizophrenia”)

Answer: Thanks to reviewer 4 for this comment. We totally agree with the reviewer. We have modified the aim of the study to adjust it to the study performed and therefore to the content of the manuscript.  

I have two main concerns about the manuscript:

  1. For a paper whose stated aim is methodology development, there is very little in the manuscript about development of the methodology – essentially use of one part of an existing tool, and one additional questionnaire concerning the type and time commitment of clinicians involved in service provision. There does not appear to have been any attempt at validation either of the modified DESDE-LTC or of the additional instrument, as deployed here.

Answer: Thanks to reviewer 4 for this comment. As stated above, we have modified the aim of the study to adjust it to the study performed and therefore to the content of the manuscript.  

 While a large part of the methodology section describes the section of the DESDE-LTC that was used, there is minimal information on how it was used: merely that it was provided to researchers in the three countries, and where those researchers lacked information they contacted planners and directors of mental health services in their country. The researchers in Spain also had a previously completed DESDE-LTC as a resource.

Answer: Thanks to reviewer 4 for this comment. We agree with the reviewer and we have extended the methodology section in order to explain better the instrument and the procedures for collecting the information.

However, there is no information on how many researchers completed the questionnaire in each country, who they were (e.g. profession(s), seniority, clinical experience), the extent to which they needed to consult to acquire data in practice, whether there was any confirmation or cross-checking of the data etc. For example, from what is described, it appears possible that a single person who was confident in his/her knowledge (and therefore did not consult) but who was actually out of date or mistaken could have supplied multiple incorrect data points for a country without correction. For Hungary and Israel, the institutions named for the researchers in the methodology section appear to be primarily education and research institutions; had those researchers any direct, current clinical knowledge/experience?

Answer: Thanks to reviewer 4 for this comment. The questionnaires have been filled out by a group of clinicians and researchers from each institution, not just one person. In the case of doubts, they checked with the service planners in their area. The three centres deliver public care so they are aware of the existing care services. In the previous response to reviewer 4 we have explained this point better to make it clearer.

For the additional questionnaire, even less information has been provided. It is simply stated (L168-9) that, “Additionally each country completed a questionnaire describing, for each service, which clinicians were involved and the estimated time dedicated for them in an ordinary visit.” Was this completed by the same researchers? Was any consultation involved? Similar questions arise as for the DESDE-LTC component above.

Answer: Yes, all questionnaires were filled out by a group of clinicians and researchers from each institution. We have added a paragraph in the methods section regarding this point. 

  1. This leads to my second key concern, which is that data has been specified with a level of confidence and precision that seems very unlikely to be justifiable. This particularly the case for the data in Table 2 (the caption for which makes no mention of estimation). No objective basis for the personnel and time estimates has been provided, and it is unknown whether or how much consultation took place in their generation, yet specific time ranges (with an implied accuracy of minutes) are set out for multiple professions and settings.

Answer: Thanks to reviewer 4 for this comment. We totally agree with the reviewer. The data in table two are not objectively recorded, but are based on the experience of the researchers and clinicians at each centre. It should be noted that they are professionals with long experience in healthcare. Mention is made in the Methods section.Moreover we have included in the text the term “estimation” in order to clarify the procedure.“Additionally, professionals involved in the B section of the DESDE-LTC in each country completed a questionnaire describing each service and which type of clinicians were involved. Moreover, an estimated time dedicated for patients in each service was performed.”

For the DESDE-LTC data it is stated that for Hungary and Israel, “as there are no governmental documents covering this information, and based on the knowledge that clinical practices are homogenous throughout both countries, all data were based on the common practice at Gertner Institute and Semmelweis.” It is not clear whether the same approach was taken for the personnel and time estimates but there is nothing to suggest otherwise.

Answer: The referee is right; the information from Hungary and Israel was based on the common practice and experience of the researchers.

Even if it is accepted that all institutions offered the same range of services (and this seems implausible, given likely differences in size and budgets), it stretches credibility to assume that the same can be said of their specific operating systems to the extent that estimated patient care times for multiple professions (of unknown derivation) in one institution could be assumed to hold true nationally. In the section on limitations the possibility that data from Catalonia may not be representative of nationwide data is considered, but in fact this is true for all three countries and its likelihood places it beyond being a mere limitation; it is a significant methodological flaw that undermines the credibility of the data, especially those in Table 2. At best, they can be considered institutional estimates, and even then there appear to have been no objective quality control measures to provide any assurance of their accuracy.

Answer: Thanks to reviewer 4 for this comment: We agree with the reviewer that this study has its limitations. However, it allows us to compare the availability of services in different regions and is an example for future studies where ICT solutions have to be implemented in different countries and where the availability of mental health services in those countries has to be evaluated.

Nevertheless, the limitations paragraph has been rewritten.

Other comments:

Figure 1: The green and red squares have been defined as ‘yes’ and ‘no’ respectively; however there is no indication of what the black squares represent.

Answer: Black squares are not related to Health Care, and this is the reason why they have not been evaluated. We have added a sentence regarding this point in the footnote of tables 1.

Table 2: Based on Table 1, there are additional services in the self-help and volunteer care domain that were common to all three locations: S1.1 and S2.1. These have not been included in Table 2, although in the discussion (L207-215) it is stated that both of these involve graduate professionals at proportions below and above 60% of total full time equivalent personnel respectively. Either they should have been included in Table 2 or their absence from it should have been explained.

Answer: The reviewer is right.  We have not included information from the self-help and volunteer care domain due to the fact that this type of services are self-help and voluntary and the proportion on professionals is lower than 60%. We have not collected information from non-professional staff, as it is no relevant for the study purpose. We have added a sentence regarding this point.

From a language perspective, the manuscript is generally well written, with some minor lapses. Formatting also needs some attention, e.g. in several places, spaces between words appear to have been omitted.

Answer: Thanks to reviewer 2 for this comment. The document has been thoroughly revised by a native English speaker and changes have been made throughout the manuscript.
